## PRImary care Management of lower Urinary tract Symptoms in men: protocol for development and validation of a diagnostic and clinical decision support tool (the PriMUS study)

Bethan Pell ,[1] Emma Thomas-Jones,[1] Alison Bray,[2,3] Ridhi Agarwal,[4] Haroon Ahmed,[5] A Joy Allen,[6] Samantha Clarke,[7] Jonathan J Deeks,[8] Marcus Drake,[7] Michael Drinnan,[2,3] Calie Dyer,[1] Kerenza Hood,[1] Natalie Joseph-Williams,[5] Lucy Marsh,[1] Sarah Milosevic,[1] Robert Pickard,[9] Tom Schatzberger,[10] Yemisi Takwoingi,[4] Chris Harding,[9] Adrian Edwards[5]

For numbered affiliations see end of article.

**Correspondence to**
Bethan Pell; pellb@cardiff.ac.uk

## ABSTRACT

**Introduction** Lower urinary tract symptoms (LUTS) is a bothersome condition affecting older men which can lead to poor quality of life. General practitioners (GPs) currently have no easily available assessment tools to help effectively diagnose causes of LUTS and aid discussion of treatment with patients. Men are frequently referred to urology specialists who often recommend treatments that could have been initiated in primary care. GP access to simple, accurate tests and clinician decision tools are needed to facilitate accurate and effective patient management of LUTS in primary care.

**Methods and analysis** PRImary care Management of lower Urinary tract Symptoms (PriMUS) is a prospective diagnostic accuracy study based in primary care. The study will determine which of a number of index tests used in combination best predict three urodynamic observations in men who present to their GP with LUTS. These are detrusor overactivity, bladder outlet obstruction and/or detrusor underactivity. Two cohorts of participants, one for development of the prototype diagnostic tool and other for validation, will undergo a series of simple index tests and the invasive reference standard (invasive urodynamics). We will develop and validate three diagnostic prediction models based on each condition and then combine them with management recommendations to form a clinical decision support tool.

**Ethics and dissemination** Ethics approval is from the Wales Research Ethics Committee 6. Findings will be disseminated through peer-reviewed journals and conferences, and results will be of interest to professional and patient stakeholders.

**Trial registration number** ISRCTN10327305.

### Strengths and limitations of this study

► Prospective, multicentre study in an appropriate population in primary care.
► The index tests are tests that can be done routinely in primary care or at home by patients.
► The diagnostic models developed will be validated in a separate cohort of men from the same population.
► The assumed prevalence of the three target conditions may be different in practice.
► Some test results may be missing or difficult to obtain.

lead to poor quality of life. Three common causes of LUTS are instability of the bladder muscle (detrusor overactivity (DO)), benign enlargement of the prostate gland causing bladder outlet obstruction (BOO) and weakness of the bladder muscle (detrusor underactivity (DU)). These may be present individually or in combination.

The reference standard test for investigation of LUTS, and thus diagnosis of DO, BOO and DU, is invasive urodynamics, which takes place in secondary care. Invasive urodynamics will be conducted rather than video urodynamics, which is in line with most contemporary national and international guidelines, and is sufficient to diagnose DO, BOO and DU, to which most non-complicated adult male LUTS can be attributed. It involves insertion of catheters into the patient's bladder and rectum so that the behaviour of the bladder and outlet can be examined during filling and voiding. Owing to availability, complexity and cost, management decisions for men with LUTS are usually

## INTRODUCTION

Lower urinary tract symptoms (LUTS), such as frequent urination, a slow stream, and having to wake in the night to urinate, affect a significant proportion of older men and can

based on results from a combination of non-invasive and minimally invasive investigations instead. These include digital rectal examination (DRE) to assess prostate size, symptoms questionnaires, uroflowmetry and measurement of post void residual.

National Institute for Health and Clinical Excellence (NICE) guidelines suggest that many men referred to specialist care with LUTS are eventually managed conservatively, and so they could have remained within primary care. Male LUTS account for around four presentations per month in an average-sized general practitioner (GP) practice. This rate of presentation, although high enough to represent a large burden on the National Health Service (NHS), makes it difficult for GPs to gain sufficient expertise to be confident about diagnosis and management. Furthermore, GPs do not have access to simple tools giving an indication of the most likely cause of symptoms to guide treatment and management. Making such a tool available should improve treatment efficacy, standardise treatment, reduce unnecessary referrals, expedite referral of those requiring specialist care and thus improve cost-effectiveness of NHS care.

This led to the National Institute for Health Research (NIHR) releasing a 2015 health technology assessment (HTA) commissioned call (HTA number 15/40) seeking the delivery of: 'The development of a decision aid to help inform the choice of treatment or need for specialist referral for men presenting with lower urinary tract symptoms in primary care'. Our team was successful in obtaining this funding and here we describe the protocol for our study: 'Primary care management of lower urinary tract symptoms in men: Development and validation of a diagnostic and clinical decision support tool' (the PRImary care Management of lower Urinary tract Symptoms (PriMUS) study).

### Aims, objectives and outcome measures

The PriMUS study aims to develop three diagnostic prediction models based on the results of simple clinical tests that can provide a useful prediction of urodynamic observations in men with LUTS. We will assess the diagnostic accuracy of these models, which will be implemented in software, along with management recommendation algorithms to form a clinical decision support tool for use in UK primary care. Our primary and secondary objectives and measures are outlined as follows.

### Primary objectives
► Develop a statistical model to predict the likelihood of three urological conditions (BOO, DO and DU) based on a series of non-invasive index tests, with invasive urodynamics as the reference standard.
► Estimate the diagnostic accuracy of the above statistical model in an independent validation cohort.

### Secondary objectives
► Develop a series of patient management recommendations and thresholds for clinically useful diagnostic

prediction by expert consensus and with reference to current clinical guidelines that map to the diagnoses predicted by the statistical model.
► Combine the statistical model and management recommendations into an online tool that will form the prototype clinical decision support tool.
► Complete a qualitative study to explore the feasibility of introducing the clinical decision support tool into primary care including potential acceptability to primary care staff and patients.
► Collect NHS costs involved in delivering the new pathway and compare with cost of standard pathway calculated from NHS and other sources.

### Primary outcome
Sensitivity and specificity of the diagnostic models for detecting DU, BOO and DO will be determined. The three conditions will be coded as binary outcomes (present/absent).

### Secondary outcomes
► A patient management algorithm to guide initial treatment for men with LUTS.
► A prototype online clinical decision support tool for use in primary care.
► Qualitative summary of patients' and clinicians' views on the use of a LUTS clinical decision support tool in the primary care setting.
► Costs/savings of implementation of the primary care LUTS clinical decision support tool both from a population and individual patient perspective.

## METHODS AND ANALYSIS
### Study design
This is a prospective, diagnostic accuracy study involving the development and validation of a diagnostic tool. An internal pilot will assess primary care recruitment, acceptability of the reference test (invasive urodynamics) and data collection. Two cohorts of participants, one for development of the prototype diagnostic tool and other for validation, will undergo a series of index tests (see table 1) and the invasive reference standard (urodynamics) in approximately 90 GP practices across Newcastle upon Tyne, Wales and Bristol (a list of study sites can be found on ISRCTN). There will also be qualitative data collection to explore acceptability of the urodynamics test and develop management recommendations for the tool and for user-testing of the prototype (see table 2).

### Participants
Adult men who consult their GP with one or more LUTS in UK primary care settings.

### Inclusion criteria
► Men aged 16 years and over.
► Men who present to their GP with a complaint of one or more bothersome LUTS (this includes men on current treatment, but who are still symptomatic).

**Table 1** Index tests and input parameters that will be tested for use in the three logistic regression models

| Test | Result | Input parameters that will be tested for use in the three logistic regression models (result or unit) |
|---|---|---|
| Relevant demographics | Age in years | Age (years) |
| Physical examination of abdomen | Bladder palpable/not palpable | N/A |
| Digital rectal examination | Prostate mild/moderate/severe enlargement Further assessment for prostate cancer required/not required | Prostate size (enlarged/not enlarged) |
| Prostate Specfic Antigen (PSA) | PSA value—established thresholds for further assessment for prostate cancer (typically >3 ng/mL) or benign enlargement (typically ≥1.5 ng/mL). For clinical decision support tool: continuous variable in ng/mL | PSA (ng/mL) |
| International Consultation on Incontinence Questionnaire Male Lower Urinary Tract Symptoms Short Form | Total score (0–52), voiding symptom score (0–20), storage symptom score (0–24). Individual symptom bother scores scored separately from symptom severity scores (0–130) | Storage/incontinence symptoms subscore Voiding symptoms subscore |
| International Prostate Symptom Score questionnaires | Total score (0–35) | Storage/incontinence symptoms subscore Voiding symptoms subscore |
| Bladder diary | Waking (day) time frequency, sleeping (night) time frequency, 24 hours voided volume, daytime voided volume, nocturnal voided volume, average volume voided each void, total urgency scores | Mean urgency score Mean 24 hours fluid intake (mL) |
| Uroflowmetry (Flowtaker) | Maximum flow rate, voided volume against normal age-adjusted range. Single value in mL/s | Median maximum flow rate (mL/s) Median voided volume (mL) Mean 24 hours frequency Mean nocturia |
| Post void residual | Residual volume against normal age-adjusted range. Single value in mL | Post void residual volume (mL) |

PSA, prostate-specific antigen.

▶ Men able and willing to give informed consent for participation in study.
▶ Men able and willing to undergo all index tests and reference test, and who complete study documentation.

### Exclusion criteria
▶ Men with neurological disease or injury affecting lower urinary tract function.
▶ Men with LUTS considered secondary to current or past invasive treatment or radiotherapy for pelvic disease.
▶ Men with contraindications to urodynamics such as heart valve or joint replacement surgery within the last 3 months, or immunocompromised/immunosuppressed men.
▶ Men with indwelling urinary catheters or who carry out intermittent self-catheterisation.
▶ Men whose initial assessment suggests that clinical findings are suggestive of:
  – Prostate or bladder cancer according to standard NHS cancer pathways. If later deemed unlikely, they become eligible for study participation.

  – Recurrent or persistent symptomatic urinary tract infection (UTI). If UTI is successfully treated but LUTS remain, they become eligible for study participation.
  – Urinary retention, for example, palpable bladder after voiding.
▶ Men unable to consent in English or Welsh where a suitable translator is not available.

### Test selection
Test selection, including the reference standard, was informed by a systematic review included in the relevant NICE guideline CG97[1] updated with a study-specific unpublished selective review by our group in 2015, the judgement of the expert clinical members of the study team and the stipulations of the funding commissioning brief. All participants undergo all tests that are a combination of:
▶ Tests carried out for eligibility assessment prior to enrolment, as described earlier.
▶ Tests carried out at participant visits to a primary or secondary care location for the purpose of the study following enrolment.

**Table 2** Secondary substudies

| Qualitative studies | Details |
| --- | --- |
| **Patient and Clinician Acceptability Interviews—Internal Pilot Phase** | During the pilot phase, we will conduct semistructured qualitative interviews (n=30–40) with patients (consenting and declining entry to the main study) and participating clinicians to assess the acceptability of the urodynamic procedure and the PriMUS study, as part of our progression criteria. Interview schedules will be developed in discussion with clinician and patient representatives of the Study Management Group. Interview guides will broadly explore: practicality and acceptability of conducting urodynamic investigations (and experiences of patient participants); reactions to and experiences of the study processes (including barriers/facilitators). An iterative approach will be taken, so that schedules can be refined to further explore unanticipated themes that arise during data collection. Interview transcripts will be entered into NVivo qualitative analysis software and analysed using Framework Analysis (using key topic areas as the framework).[9] Data will be used to inform strategies that will maximise recruitment and retention. |
| Development of Management Recommendations | Algorithms are required to link outputs from the statistical models, which will be likelihoods of each target condition, with patient management recommendations to form the clinical decision support tool. The starting point will be recommendations from the relevant NICE clinical guideline. Qualitative work with urologists will support the development of these management recommendations, through posing a range of clinical case scenarios to urologists using interview and questionnaire methodologies (n=15–20). Urologists will be asked to how they would manage these scenarios, with a focus on thresholds for treatment and strategies for multiple diagnoses. |
| Tool Feasibility Assessment | The aim of the user-testing phase will be to build on the interviews conducted as part of the pilot phase evaluation to assess GPs' attitudes and reactions to the prototype clinical decision support tool. GPs (n=10–12) will be sent the tool prior to the semistructured telephone interview and asked to use it. The interview schedule will explore the following: ease of use, content, design and perceived acceptability and feasibility of using the tool in routine primary care settings (allowing succinct exploration of the prototype tool). Interview transcripts will be entered into NVivo qualitative analysis software and analysed using Framework Analysis (using key topic areas as the framework).[9] Feedback will be used to improve and refine the tool. |

GPs, general practitioners; NICE, National Institute for Health and Clinical Excellence; PriMUS, PRImary care Management of lower Urinary tract Symptoms.

► Tests carried out by the participant at home following enrolment.

### Index tests
Twelve potential parameters will be considered for the three logistic regression models. The investigations that provide these parameters are described in table 1.

### Reference standard
Our reference standard is invasive urodynamics, a test routinely carried out in a specialist care setting for the investigation of LUTS. For this study, it will be performed using portable equipment (Goby, Laborie, Mississauga, Canada) in either a primary or secondary care location, by specially trained urodynamic nurses, according to International Continence Society standards.[2] Safety information is covered in the Safety and pharmacovigilance section.

### Study procedures
#### Data collection
GPs, primary care nurses or an appropriately trained delegate will undertake the data collection related to all the index tests. Specialist-trained urodynamic nurses will undertake the data collection for the reference test.

#### Data management
All data collection will be done by electronic data capture using a bespoke database developed by the Cardiff University Centre for Trials Research Clinical Trials Unit (CTR), and paper copies of all case report forms will be available.

#### Identification and screening
All men will be identified either opportunistically during a GP consultation or by regular, predefined primary care database searches. They must undergo three screening tests prior to enrolment into the study: a physical examination of the abdomen (palpable bladder check), DRE and prostate-specific antigen test. The latter two test results are accepted if they have undergone these investigations within the last 6 months.

#### Informed consent
Informed consent will be obtained in the first study visit (Study Visit Part A) prior to any study procedures by those suitably trained and on the delegation log. Eligible patients will be given time to consider before being asked to sign the consent form. Once consented, participants will be allocated a unique study number (participant ID).

Separate informed consent will be taken for participation in the qualitative data collection.

#### Withdrawal
Patients will be notified that they can withdraw consent for their participation in the study at any time during the study period.

#### Study Visit Part A
Once informed consent is obtained, the remaining index tests will be collected. These include a baseline assessment (collecting demographic information, relevant medication and medical history) and two self-reported

questionnaires: International Prostate Symptom Score and International Consultation on Incontinence Questionnaire. Participants will be given the bladder diary to complete for 3 days at home and instructed to bring it during their invasive urodynamic visit (Study Visit Part B).

### Study Visit Part B—reference standard

On arrival, the patient will be asked to pass urine into a flowmeter in private, after which a measurement of post void residual ultrasound (one of the index tests; see later) will be made. A dual lumen catheter (one channel to fill the bladder and the other to measure intravesical pressure, $P_{ves}$) will be inserted into the bladder via the urethra, and a single lumen catheter will be inserted into the rectum to measure abdominal pressure ($P_{abd}$). Detrusor pressure ($P_{det}$), generated by the bladder muscle itself, is calculated by subtracting $P_{abd}$ from $P_{ves}$.

### Filling phase

The patient will be asked to bring their completed bladder diary (one of the index tests; see earlier) to their urodynamics appointment, providing the urodynamic nurse with an indication of their maximum bladder capacity. The patient's bladder will be filled with sterile saline at a maximum rate of 50 mL/min. They will be asked to report the first sensation of bladder filling, followed by the point at which they feel the normal desire to void and finally the strong desire to void. At this point, bladder filling will be stopped and provocation, in the form of running taps and asking the patient to cough, will be performed.

### Voiding phase

Following provocation, the patient will be given permission to void, marking the start of the voiding phase. Voided volume ($V_{void}$) and flow rate (Q) will be measured as they pass urine into the flowmeter.

If either the filled or voided volumes are below 150 mL, the filling and voiding phases will be repeated once more using a maximum filling rate of 20 mL/min.

### Diagnostic definitions

Definition of our three target conditions will be based on the following parameters measured during invasive urodynamics and subsequently read from a graphical representation of the test:
1. Maximum detrusor contraction pressure during the filling phase.
2. Maximum flow rate during the voiding phase ($Q_{max}$).
3. Detrusor pressure at the point of maximum flow rate ($P_{detQmax}$).

If there are no detrusor contractions during filling, DO is not present. If there are any contractions (contraction pressure >0), DO is present.

Diagnosis of BOO is based on the BOO index (BOOI), defined as $P_{detQmax}-2*Q_{max}$. BOO is present if BOOI >40 and absent if BOOI ≤40.

Diagnosis of DU is based on the bladder contractility index (BCI), defined as $P_{detQmax}+5*Q_{max}$. DU is present if BCI<100 and absent if BCI≥100.

### Debrief process and monitoring process

The urodynamic nurses will debrief the patient following the urodynamic procedure providing them with a post-urodynamics leaflet and safety card. The urodynamic nurse will also instruct the patient that they will receive a 3-day follow-up phone call to monitor for any related adverse events.

### 3-day follow-up phone call

The urodynamic nurses will contact the patient 3 days (with flexibility if phone call falls on weekend) after their urodynamic procedure to monitor for any adverse events and serious adverse events (SAEs). Any SAEs are subsequently recorded by the urodynamic nurses and processed centrally by CTR. This process is outlined in the Safety and pharmacovigilance section (see later).

### Review process

Invasive urodynamics is a complex investigation and interpretation can be challenging. Furthermore, since standard practice involves interaction between the reference and some index tests as described earlier, their primary interpretation in this study is not blinded. Therefore, a review process will be implemented to ensure the integrity of the reference standard. All studies will be second-read by a blinded reviewer to extract the three parameters described earlier. If any of the resulting diagnoses differ between the nurse and this reviewer, the case will go to a second non-blinded reviewer who makes the final decision.

### Uroflowmetry* (Flowtaker)

The patient is provided with the Flowtaker at the end of their invasive urodynamic visit. They will be provided with an information sheet on how to use this and instructed not to start this until given the green light to do so during their 3-day follow-up phone call. The flowmeter will be given with a prepaid envelope for the patient to post back to their urodynamic nurse for data upload.

### GP summary report

Once the reference and index tests have taken place, results are compiled into a report which is provided to GPs, along with a summary of relevant NICE-recommended managements,[1] to help inform management of the patient.

### 6-month follow-up

A review of the patient's medical notes will take place 6 months after the patient's treatment and management decision with the GP. This will include any changes to treatment or management and whether they have since been referred to secondary care. Figure 1 depicts a flow diagram of the patient pathway.

### Patient and Public Involvement

Patient and Public Involvement (PPI) will be sought throughout the research process, from conceptualisation to dissemination. PPI will contribute to the design, set

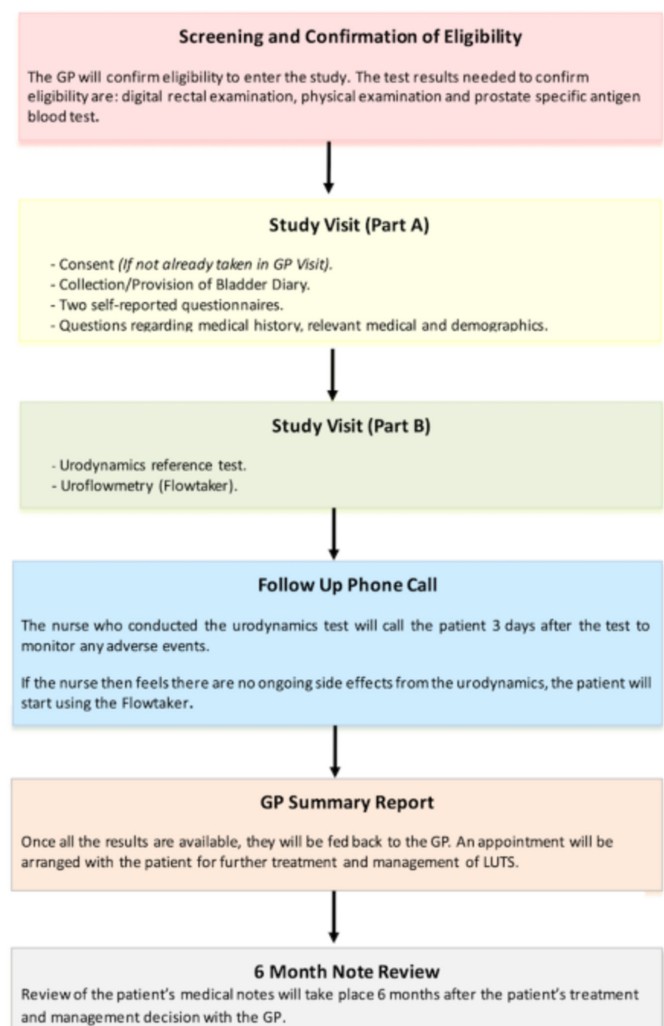

**Screening and Confirmation of Eligibility**

The GP will confirm eligibility to enter the study. The test results needed to confirm eligibility are: digital rectal examination, physical examination and prostate specific antigen blood test.

**Study Visit (Part A)**

- Consent (If not already taken in GP Visit).
- Collection/Provision of Bladder Diary.
- Two self-reported questionnaires.
- Questions regarding medical history, relevant medical and demographics.

**Study Visit (Part B)**

- Urodynamics reference test.
- Uroflowmetry (Flowtaker).

**Follow Up Phone Call**

The nurse who conducted the urodynamics test will call the patient 3 days after the test to monitor any adverse events.

If the nurse then feels there are no ongoing side effects from the urodynamics, the patient will start using the Flowtaker.

**GP Summary Report**

Once all the results are available, they will be fed back to the GP. An appointment will be arranged with the patient for further treatment and management of LUTS.

**6 Month Note Review**

Review of the patient's medical notes will take place 6 months after the patient's treatment and management decision with the GP.

**Figure 1** Flow diagram depicting the patient pathway in the PriMUS study. GP, general practitioner; LUTS, lower urinary tract symptoms; PriMUS, PRImary care Management of lower Urinary tract Symptoms.

up and management of thestudy as well as the progress and conduct of the study as part of studysub-groups. PPI representatives will also lead on dissemination activities tostudy participants and the wider public.

### Safety and pharmacovigilance

Invasive urodynamics has the potential to cause adverse events. A medical doctor will be required on-site while the test is taking place. Due to a 5% risk of UTI,[3] the urodynamic nurse will also provide the patients with a post-urodynamics debrief sheet following the test, informing them on the importance of drinking plenty of water for 24 hours following the test, how to identify signs of an infection and to seek medical care if they suspect they have one.

Adverse events will be captured by the urodynamic nurses, either during Study Visit Part B or during the 3-day follow-up phone call. For SAEs, an assessment of causality between the event and the study intervention, and the expectedness of the event, will be carried out

by the Principle Investigator, or delegated urodynamic nurse, and then independently by a clinical reviewer. If the clinical reviewer classifies the event as *probably* or *definitely* caused by the intervention, it will be classified as a serious adverse reaction.

### Sample size

Sample size calculations were carried out separately for the model development and validation cohorts. For both, we used estimated prevalences for DO, BOO and DU of 57%, 31% and 16%, respectively, based on previous literature[4 5] and clinical expertise.

### Development cohort

The sample size for developing our predictive models was based on a rule of thumb suggesting that five events per variable are required.[6] We chose a sample size of 350 to allow at least 11 variables in each model. This was driven by our lowest estimated prevalence of 16% for DU, giving 56 events (DU diagnoses).

### Validation cohort

The sample size for the validation cohort was chosen to ensure that estimates of test accuracy are made with adequate precision. We deem sensitivity and specificity of 75% to be the minimum clinically useful performance. We chose a sample size of 325, giving estimates of sensitivity of 75% to within 8%, 10% and 14% for DO, BOO and DU respectively, based on 'positive' samples of 185, 101 and 52, and estimates of specificity of 75% to within 8%, 7% and 6%, based on 'negative' samples of 140, 224 and 273. Better sensitivity and specificity will give narrower CIs.

### Attrition

To allow an attrition rate of 20%–25%, the resulting total of 675 was increased to give a final sample size of 880.

### STATISTICAL ANALYSIS

Model development will be performed using results from the first 350 data sets, and external model validation performed using the subsequent 325 data sets.

### Model development

Candidate predictor variables will be selected from those listed in table 1. Their selection has been informed by subject knowledge using literature review and expert judgement. As predictor distributions should be wide to facilitate reliable predictions, we will explore the distribution of each predictor prior to selection. Relationships between predictors will also be investigated; where indicated we will group related variables into a composite variable or exclude if highly correlated with other variables. Candidate predictors will not be selected based on univariable analyses; this practice is discouraged because predictors that may be important in a multivariable model can be missed and may also lead to overoptimistic models. Therefore, all selected candidate predictor variables will be included in the multivariable

logistic regression models without evaluation of association between outcome and predictor and assessment of statistical significance. To gain maximum diagnostic information, continuous variables will not be categorised. We will allow for non-linearity by using a multivariable fractional polynomial approach to identify appropriate transformations. This may lead to the inclusion of non-linear terms in the models, thus increasing the number of variables in the models. Using multiply imputed data and Rubin's rule, we will develop each model using backward elimination with a p value of 0.157 to select predictors for inclusion in each model. We chose this p value because it is known to be a good proxy for the Akaike information criterion approach. If the repeated use of Rubin's rule is computationally challenging, we will use the approximation to Rubin's rule recommended by Wood et al.[7]

## Model validation

The predictive performance of each model will be assessed in terms of discrimination, that is the ability to distinguish between those who do or do not have a particular diagnosis, and calibration meaning agreement between predicted and observed probabilities. Discriminative ability will be assessed using the c-index and its 95% CIs. For a logistic model, this is equivalent to the area under the Receiver Operator Characteristic (ROC) curve. Calibration will be evaluated in two ways. Calibration plots of average observed probability against predicted probability will be used to visually assess calibration. Within each quintile or decile of predicted probability (depending on the distribution of data), the average predicted probability will be compared with the corresponding observed proportions. We will also quantify calibration by estimating the calibration slope of the prognostic index (linear predictor) using logistic regression with the linear predictor as the covariate.

The apparent c-index and calibration slope will be estimated for each model. Bootstrapping will be used for internal validation to assess model overfitting and optimism. For each model, we will obtain 100 bootstrap samples from each imputed data set and repeat the variable selection process. The optimism is the difference between the c-index from the bootstrap sample and that from the original imputed data set. The average optimism will be determined across bootstrap samples and imputed data sets, and the optimism-adjusted c-index will be calculated by subtracting the average optimism from the apparent c-index of the original model. Similarly, we will obtain the optimism-adjusted calibration slope. The optimism-adjusted calibration slope will be used as the uniform shrinkage factor to correct a model.

We will externally validate the models and calculate performance statistics (c-index and calibration slope) using the validation cohort. The value for the calibration slope should ideally be one signifying perfect agreement between the predicted probabilities and the observed probabilities. A calibration slope <1 indicates that a model overpredicts while a calibration slope >1 indicates

underprediction. From the qualitative research, we will ascertain distributions of probability (risk) thresholds for clinical usefulness of the prediction in guiding treatment of each condition. The sensitivity and specificity (and their corresponding 95% CIs) will be calculated for these risk thresholds and plotted on an ROC plot for each model.

## Missing data

Patterns of missing data will be investigated to infer the ease with which each parameter can be obtained in practice. In the event of missing data, multiple imputation by chained equations[8] will be used to impute missing values to avoid bias and make best use of the data.

## SECONDARY SUBSTUDIES
### Study management

The study is sponsored by Cardiff University, coordinated by CTR and co-led by The Newcastle upon Tyne Hospitals NHS Foundation Trust. The other partner organisations will be Birmingham University, University of Bristol and North of England Commissioning Support.

### Study Management Group

The Study Management Group (SMG) will meet monthly throughout the course of the study and will include the chief investigators, co-applicants, collaborators, study manager, data manager and administrator. Two patient representatives will also attend and contribute to the conception, design and management of the study, as well as patient-facing materials. SMG members will be required to sign up to the remit and conditions as set out in the SMG Charter.

### Study Steering Committee

An independent Study Steering Committee (SSC) consisting of an independent chairperson, two independent members and two patient representatives will provide oversight of the PriMUS study. Instead of a separate Independent Data Monitoring Committee, the SSC will also provide oversight of all matters related to patient safety and data quality. Members will be required to sign up to the remit and conditions as set out in the SSC charters and will meet annually.

## ETHICS AND DISSEMINATION
### Research approvals

The Wales REC 6 has approved the study (17/WA/0155) on 20 June 2017 and subsequent R&D approval for Wales on 21 August 2017 and HRA approval on 23 August 2017.

All study participants will give informed consent before taking part (see earlier).

The following substantial amendments were made to the trial and were communicated to all trial sites:

Substantial Amendment 1 (3 October 2017); Substantial Amendment 2 (10 January 2018); Substantial Amendment 3 (20 April 2018); Substantial Amendment 4 (26

February 2019); Substantial Amendment 5 (6 June 2019); Substantial Amendment 6 (6 September 2019).

## Dissemination plan

Following completion of the study, a final report will be prepared for the NIHR Journal series. A paper describing the primary results will be submitted to a high impact, international, peer-reviewed journal. Qualitative studies and substudies will also be submitted for publication. We will present our findings at national and international scientific meetings.

With the assistance of our collaborators and lay representatives, we will disseminate the study findings to a wide NHS and general audience and promote uptake of outputs into clinical care. This will include presentations at meetings and written executive summaries for key stakeholder groups such as Primary Care Trusts, Secondary Care Trusts, Health Boards, Royal Colleges, Medical Schools and relevant patient groups.

All publications and presentations related to the study will be authorised by the SMG in accordance with the study's publication policy.

**Author affiliations**
[1]Centre for Trials Research, Cardiff University, Cardiff, UK
[2]Medical Physics, Newcastle upon Tyne Hospitals NHS Foundation Trust, Newcastle upon Tyne, UK
[3]Translational and Clinical Research Institute, Newcastle upon Tyne Hospitals NHS Foundation Trust, Newcastle upon Tyne, UK
[4]Test Evaluation Research Group, Institute of Applied Health Research, University of Birmingham, Birmingham, UK
[5]Division of Population Medicine, Cardiff University, Cardiff, UK
[6]NIHR In Vitro Diagnostics Co-operative, Newcastle University, Newcastle upon Tyne, UK
[7]North Bristol NHS Trust, Westbury on Trym, UK
[8]Public Health, Epidemiology and Biostatistics, University of Birmingham, Birmingham, UK
[9]Department of Urology, Newcastle upon Tyne Hospitals NHS Foundation Trust, Newcastle upon Tyne, UK
[10]Corbridge Health Centre, NHS Northumberland Clinical Commissioning Group, Newcastle, UK

**Acknowledgements** The authors would like to thank Professor Robert Pickard (deceased 24 July 2018) for development of the research question, study design, obtaining the funding and implementation of the study protocol. They would also like to thank the research nurses for their support during the study: Debra Barnett, Laura Bevan, Jane Davies, Alison Edwards, Gareth Kennard-Holden, Lisa Mellish, Joanne Sullivan and Joanne Thompson. The Centre for Trials Research receives funding from Health and Care Research Wales and Cancer Research, UK. AJA is supported by the National Institute for Health Research (NIHR) Newcastle In Vitro Diagnostics Co-operative. The views expressed are those of the authors and not necessarily those of the NIHR or the Department of Health and Social Care. They would also like to acknowledge the contribution of the Trial Steering Committee members, namely Professor Tom Fahey, Professor Rafael Perea, Dr Gail Hayward, Dr Ian Pearce and Mr Alan Pryce.

**Contributors** AE and CH are co-chief investigators of this study. AE and CH, along with HA, AJA, AB, MaD, JJD, MiD, KH, NJ-W, RP, TS, YT and ET-J, led the development of the research question, study design, obtaining the funding and implementation of the study protocol. BP is the study manager and ET-J is the senior study manager who coordinated the operational delivery of the study protocol and recruitment. SC provides research nurse insight and support. NJ-W and SM are the qualitative researchers. YT and RA are the study statisticians. CD and LM are the data managers. All authors listed provided critical review and final approval of the manuscript.

**Funding** This study is funded by the NIHR (Health Technology Assessment programme), funder reference 15-40-05. Cardiff University, Research and Innovation Services Department, Contracts Team, Cardiff University, 30-36 Newport Road, Cardiff, CF24 0DE. Contact person: Ms Helen Falconer; FalconerHE@cardiff.ac.uk. Sponsor reference: SPON 1553-16.

**Disclaimer** The views expressed are those of the authors and not necessarily those of the NHS, the NIHR or the Department of Health and Social Care. Neither the Sponsor nor the Funder had any role on the study design; collection, management, analysis and interpretation of data; writing of this manuscript or in the decision to submit this manuscript for publication.

**Competing interests** One of the index tests, Flowtaker, was developed by a team from Newcastle upon Tyne Hospitals (NuTH) and Newcastle University, including two individuals who are grant co-applicants, members of the study management team and co-authors (AB and MiD). In 2014, the device was licensed to MMS (Enschede, the Netherlands) and royalties from the sale of the device were paid to NuTH (not to the individuals). MMS was subsequently acquired by Laborie who removed Flowtaker from the market in January 2018.

**Patient and public involvement** Patients and/or the public were involved in the design, or conduct, or reporting, or dissemination plans of this research. Refer to the Methods section for further details.

**Patient consent for publication** Not required.

**Provenance and peer review** Not commissioned; externally peer reviewed.

**ORCID iD**
Bethan Pell http://orcid.org/0000-0002-0786-6339

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
