## [Reviewer comments · BMJ Open]

ARTICLE DETAILS

TITLE (PROVISIONAL)	Protocol for PRImary care Management of lower Urinary tract Symptoms in men: Protocol for development and validation of a diagnostic and clinical decision support tool (The PriMUS Study).
AUTHORS	Pell, Bethan; Thomas-Jones, Emma; Bray, Alison; Agarwal, Ridhi; Ahmed, Haroon; Allen, A; Clarke, Samantha; Deeks, Jonathan; Drake, Marcus; Drinnan, Michael; Dyer, Calie; Hood, Kerenza; Joseph-Williams, Natalie; Marsh, Lucy; Milosevic, Sarah; Pickard, Robert; Schatzberger, Tom; Takwoingi, Yemisi; Harding, Chris; Edwards, Adrian

VERSION 1 - REVIEW

REVIEWER	Emily Cooper Public Health England, Primary Care and Interventions Unit I work to support the maintenance and review of UTI diagnostic guidance published by Public Health England.
REVIEW RETURNED	28-Apr-2020

GENERAL COMMENTS	Really pertinent subject. I look forward to reading through the results. There needs to be further explanation on how you will differentiate or clarify the relationship between LUTS and UTI. How will you handle men with a UTI that is not a recurrent or persistent UTI? What about men with previously documented Asymptomatic Bacteriuria? Given the challenges with UTI diagnosis, cut-off parameters for urine analysis, and ASB in older men, will you link data on urine culture to the data analysis? Have you considered including urinalysis results as part of your logistic regression model? What diagnostic criteria will primary care providers use to diagnose a lower UTI in men (can be subjective in primary care)? What framework or model will you use to structure the qualitative interview schedules? Suggest that you have a patient representative as part of the study/research group, especially if any of the tools will be patient facing.
--

REVIEWER	Hann-Chorng Kuo Department of Urology, Hualien Tzu Chi General Hospital, Hualien, Taiwan
REVIEW RETURNED	07-May-2020

GENERAL COMMENTS	This is a protocol report, not a clinical study report. I cannot figure out why the journal will accept this kind of manuscript. However, this study is well designed and the study rationale is clinically relevant. Several critical points should be reconsidered in this protocol: 1) Why the age of inclusion criteria is 16 years? The study result is supposed to provide guidance for GP to diagnose and treat LUTS in adult men. 2) Is the invasive urodynamics by pressure flow study enough to be a reference standard? Why not video urodynamic study? 3) The standard of uroflowmetry should define the adequate volume voided for the obtainment of maximum flow rate.
---

VERSION 1 – AUTHOR RESPONSE

Reviewer: 1

Reviewer Name: Emily Cooper

Institution and Country: Public Health England, Primary Care and Interventions Unit, UK Please state any competing interests or state 'None declared': I work to support the maintenance and review of UTI diagnostic guidance published by Public Health England.

Really pertinent subject. I look forward to reading through the results.

1) There needs to be further explanation on how you will differentiate or clarify the relationship between LUTS and UTI. The aim of the study is not to define the relationship between LUTS and UTI. Men with clinical suspicion of a UTI (i.e. symptomatic), or persisting/recurrent UTI are excluded from the study, unless a UTI is resolved and background LUTS remain. Existing national and international urodynamic guidelines would advise that any assessment is deferred in the presence of a UTI and these have been followed in PRIMUS.

2) How will you handle men with a UTI that is not a recurrent or persistent UTI? What about men with previously documented Asymptomatic Bacteriuria? In relation to the response above, men with clinical suspicion UTI are excluded from the study. However, if a man with an acute episode of UTI is treated and no longer has UTI symptoms but does have ongoing LUTS, he is potentially eligible for the study. Given that by definition asymptomatic bacteriuria does not produce any symptoms we have not listed this as an exclusion criterion.

3) Given the challenges with UTI diagnosis, cut-off parameters for urine analysis, and ASB in older men, will you link data on urine culture to the data analysis? Have you considered including urinalysis results as part of your logistic regression model? Due to the exclusion of those with a UTI diagnosis, the only linkage of data regarding UTI in the study is via the assessment of complications of the urodynamic assessment, which participants are informed occurs in 5% of those undergoing urodynamic assessments.

4) What diagnostic criteria will primary care providers use to diagnose a lower UTI in men (can be subjective in primary care)? In line with the British Infections Association guidelines, the diagnosis of UTI by primary care providers is based on defined symptoms suggestive of UTI.

5) What framework or model will you use to structure the qualitative interview schedules?

Semi-structured interview schedules will be developed for the acceptability interviews in discussion with clinicians and patient representatives on the study management group. An iterative approach will be taken, so that schedules can be refined to further explore unanticipated themes that arise during data collection. Interview schedules for the management recommendations work will be structured so that findings can be compared with questionnaire data. Interview schedules for the tool feasibility work will also be structured in order to allow succinct exploration of key aspects of the proposed tool. We have included this information in Table 2 of the main manuscript, page 12.

6) Suggest that you have a patient representative as part of the study/research group, especially if any of the tools will be patient facing. Thank you for raising this really important point. We agree the patient representation is vital in research and can confirm that we have two patient representatives for this study, who attend SMGs and review patient-facing documentation. We have included a sentence to clarify this under Study Management page 13, line 4-5.

Reviewer: 2

Reviewer Name: Hann-Chorng Kuo

Institution and Country: Department of Urology, Hualien Tzu Chi General Hospital, Hualien, Taiwan

Please state any competing interests or state 'None declared': None declared

This is a protocol report, not a clinical study report.

I cannot figure out why the journal will accept this kind of manuscript.

However, this study is well designed and the study rationale is clinically relevant.

Several critical points should be reconsidered in this protocol:

1) Why the age of inclusion criteria is 16 years? According to the Health Research Authority, in the UK those aged 16 and over are deemed capable of giving consent on their own behalf. Therefore, although we expected the majority of participants to be older men, we opted for the lower age limit to ensure broad inclusion of participants.

2) Is the invasive urodynamics by pressure flow study enough to be a reference standard? Why not video urodynamic study? Invasive "simple" urodynamic investigation was included in the study protocol rather than video urodynamics for a variety of reasons. Firstly (and most importantly) 'simple' (non-video) urodynamic investigation is sufficient to diagnose detrusor overactivity/detrusor underactivity/bladder outlet obstruction, which are the index diagnoses in PRIMUS and are the conditions to which most non-complicated adult male LUTS can be attributed. Additional factors such as cost, resource and risk associated with video urodynamics i.e. X-rays/radiographer/radiation-safe location/use of contrast agent were also factored into the decision making process during protocol development. Finally, video urodynamic investigation is usually reserved for discrete patient populations and men presenting to primary care with lower urinary tract symptoms do not fall within the indications for a video urodynamic study. We have included a sentence to clarify this on page 4, line 9-12.

3) The standard of uroflowmetry should define the adequate volume voided for the obtainment of maximum flow rate. In line with the guidelines of the International Continence Society a valid urine flow study carried out in a clinic setting requires a voided volume of 150 mls. However, this stipulation cannot be enforced for home uroflowmetry. The median maximum flow rate from home uroflowmetry (one of the index tests) is calculated from all voids regardless of volume (i.e. voids <150 ml are not excluded). For The PriMUS Study, habitual low urine flow rates and/or habitual low volume voids from home uroflowmetry is potentially extremely helpful diagnostically and these parameters will be assessed for their predictive ability to help diagnose urological conditions in the statistical analysis.